# New Onset of Eosinophilic Granulomatosis with Polyangiitis Following mRNA-Based COVID-19 Vaccine

**DOI:** 10.3390/vaccines10050716

**Published:** 2022-05-03

**Authors:** Emanuele Nappi, Maria De Santis, Giovanni Paoletti, Corrado Pelaia, Fabrizia Terenghi, Daniela Pini, Michele Ciccarelli, Carlo Francesco Selmi, Francesca Puggioni, Giorgio Walter Canonica, Enrico Heffler

**Affiliations:** 1Department of Biomedical Sciences, Humanitas University, 20072 Pieve Emanuele, Italy; emanuele.nappi@humanitas.it (E.N.); maria.de_santis@hunimed.eu (M.D.S.); giovanni.paoletti@hunimed.eu (G.P.); carlo.selmi@hunimed.eu (C.F.S.); giorgio_walter.canonica@hunimed.eu (G.W.C.); 2IRCCS Humanitas Research Hospital, 20089 Rozzano, Italy; fabrizia.terenghi@humanitas.it (F.T.); daniela.pini@humanitas.it (D.P.); michele.ciccarelli@humanitas.it (M.C.); francesca.puggioni@humanitas.it (F.P.); 3Department of Health Sciences, University “Magna Græcia” of Catanzaro, 88100 Catanzaro, Italy; pelaia.corrado@gmail.com

**Keywords:** EGPA, ANCA-associated vasculitis, mRNA vaccine, anti-SARS-CoV-2 vaccine

## Abstract

Anti-SARS-CoV-2 vaccines are safe and effective, also in individuals with allergic and immune-mediated diseases (IMDs). There are reports suggesting that vaccines may be able to trigger de-novo or exacerbate pre-existing IMDs in predisposed individuals. Eosinophilic granulomatosis with polyangiitis (EGPA) is a small-vessel vasculitis characterized by asthma, eosinophilia, and eosinophil-rich granulomatous inflammation in various tissues. We describe the case of a 63-year-old man who experienced cardiac, pulmonary, and neurological involvement one day after the administration of the booster dose of anti-SARS-CoV-2 vaccine (mRNA-1273). A diagnosis of EGPA was made and the patient was treated with high-dose steroids and cyclophosphamide, with a good clinical response. Interestingly, our patient had experienced a significant worsening of his pre-existing asthma six months earlier, just after the first two vaccine shots with the ChAdOx1 anti-SARS-CoV-2 vaccine. It is impossible to know whether our patient would have had developed EGPA following natural SARS-CoV-2 infection or at some point in his life regardless of infectious stimuli. Nevertheless, our report may suggest that caution should be paid during the administration of additional vaccine doses in individuals who experienced an increase in IMD severity that persisted over time following previous vaccine shots.

## 1. Introduction

Anti-SARS-CoV-2 vaccines are the best tool available to hinder the progress of the COVID-19 pandemic. Although anti-SARS-CoV-2 vaccines are highly safe and effective [1,2,3], there are reports suggesting that they might be able to trigger de-novo or exacerbate pre-existing immune-mediated diseases (IMDs) [4,5,6,7]. Although whether vaccines play a role in IMD development is still debated, evidence in favor of this hypothesis was available even before the introduction of anti-SARS-CoV-2 vaccines [8,9,10]. Similar to viral infections, molecular mimicry and bystander activation have been proposed as possible mechanisms underlining the connection between vaccines and IMDs [8]. 

Eosinophilic granulomatosis with polyangiitis (EGPA) is an anti-neutrophil cytoplasmic antibody (ANCA)-associated systemic small-vessel necrotizing vasculitis characterized by eosinophil-rich granulomatous inflammation associated with asthma and eosinophilia [11]. Respiratory tract involvement is almost always present, and the peripheral nervous system, gastrointestinal tract and the myocardium are also commonly affected [11]. Hereby we describe a case of EGPA onset following the booster dose of the anti-SARS-CoV-2 vaccine mRNA-1273. 

## 2. Case Presentation

A 63-year-old man presented at the Emergency Department (ED) of our hospital for diplopia, left temporo-parietal headache, fleeting knurled scotoma, and impaired color vision, associated with a dry cough, occurring the day after the booster dose of the anti-SARS-CoV-2 vaccine mRNA-1273. The time course of clinical events is summarized in Figure 1. 

The patient’s past history was remarkable for rhinoconjunctivitis, mild asthma and allergic sensitization to house dust mites, cat dander and grass and pellitory pollens. Respiratory allergy was treated with subcutaneous allergen immunotherapy during childhood. Since then, the patient had not required asthma medication on a daily basis. 

Collecting the clinical history, it emerged that 6 months before, just after the first part of the vaccination cycle (the first two doses with the ChAdOx1 vaccine), he had developed a progressive worsening of asthma which become severe and uncontrolled despite a high dose of inhaled corticosteroids plus long-acting beta2-agonists, leukotriene receptor antagonists and tiotropium bromide, and requiring oral corticosteroid maintenance treatment. In this occasion the patient underwent a pneumological evaluation at another hospital, where the symptoms were framed as a worsening of a pre-existent mild asthma. Since then, any attempt to withdraw systemic corticosteroid was associated with asthma exacerbations.

Upon access at our ED, physical examination showed diffuse expiratory wheezes and rhonchi, bibasilar inspiratory crackles, and an adduction deficit and ptosis of the left eye, suggestive of left 3rd cranial nerve paresis. Blood tests (Figure 2) were remarkable for: leukocytosis (23.160 cells/µL) with severe hypereosinophilia (12.400 cells/µL, normal value > 500 cells/mcl), elevated serum troponin I (4030.4 ng/L, normal value < 19.8 ng/L), creatininkinase (255 U/L, normal value < 172 U/L), brain natriuretic peptide (413 pg/mL, normal value < 100 pg/mL), and C-reactive protein (2.55 mg/dL, normal value < 0.5 mg/dL). Renal and liver functions were normal, and serum ANCAs were negative. The patient underwent a chest computed tomography (CT) scan which showed bilateral pulmonary consolidations, left plural effusion and pericardial effusion. Head CT with contrast medium was normal, without evidence of vascular alterations or involvement of paranasal sinuses. Echocardiography showed pericardial effusion, altered diastolic function, preserved left ventricular ejection fraction, concentric hypertrophy of the left ventricle and a normal valvular function.

The patient was admitted to the neurology ward where he underwent additional diagnostic testing and high-dose steroid treatment with 1 g methylprednisolone boluses. Cardiac magnetic resonance imaging was compatible with eosinophilic myopericarditis, showing extensive subendocardial fibrosis (a hallmark of eosinophilic myocarditis), mild septal edema with signal enhancement at turbo inversion recovery magnitude (TIRM), T2-weighted images and increased native T2 values up to 52 milliseconds, mildly reduced left ventricular ejection fraction (49%), septal hypokinesia, and sings of active pericarditis. Nasal and laryngeal endoscopy showed nasal septum deviation without evidence of active lesions. Electromyography was relevant only for chronic L4-L5 radiculopathy. 

According to the clinical picture and laboratory test results, and considering the latest American College of Rheumatology (ACR)/European League Against Rheumatism (EULAR) classification criteria, a diagnosis of EGPA with pulmonary, neural, and cardiac involvement was made [12].

Immunosuppressive treatment with intravenous cyclophosphamide was started. Heart failure therapy with bisoprolol and enalapril was also initiated. We based our decision on whether or not to start cyclophosphamide treatment on the revised five-factor score, which considers the presence of cardiac insufficiency, renal insufficiency, gastrointestinal involvement, absence of ear, nose, and throat (ENT) manifestations, and age > 65 years; if 2 or more are present, cyclophosphamide therapy is recommended [13]. Given the presence of cardiac insufficiency and absence of ENT manifestations (Figure 3), cyclophosphamide therapy was indicated in this case.

In total, the patient received three intravenous boluses of 1 g methylprednisolone (on three consecutive days) followed by oral prednisone 75 mg/day (1 mg/kg/day), continued throughout the entire admission period. The first cycle of intravenous cyclophosphamide at a dose of 15 mg/kg was administered during hospital admission. 

After the initiation of systemic corticosteroid therapy, we observed a depletion of blood eosinophils, progressive reduction of serum troponin, resolution of the cough, and almost complete regression of 3rd cranial nerve palsy. The patient was then discharged and followed-up on an outpatient basis to continue with cyclophosphamide therapy and to taper the steroids. If the patient continues to tolerate cytotoxic therapy, our approach will be to deliver 6 monthly intravenous administrations of 15 mg/kg cyclophosphamide. The introduction of mepolizumab as a steroid-sparing strategy will be considered as soon as a durable clinical stability is achieved with the current treatment. 

## 3. Discussion

We described the clinical features of an adult male patient that developed EGPA one day after the booster dose of the anti-SARS-CoV-2 vaccine (mRNA-1273). The patient also experienced a progression from mild to severe asthma following the first two vaccine doses with the ChAdOx1 vaccine received six months previously. It is likely that the vaccination stimulus itself was able to trigger an immune response leading first to worsening of asthma, and subsequently to systemic vasculitis. The main complaint that brought the patient to the ED after the booster dose was a cranial nerve mononeuropathy, which is an atypical presentation for vasculitic neurological involvement. Instead, mononeuritis multiplex is the most common form of nervous system involvement in EGPA [14].

Our patient met the recently developed ACR/EULAR classification criteria for EGPA [12]. Although it would be hasty to speak of causality between vaccination and EGPA in our patient, the finding that he also developed severe asthma following the first two vaccine doses (six months earlier) is a strong hint towards the effects that the vaccine may have had on his immune system. EGPA is classically described as evolving in three phases: a prodromic phase characterized by respiratory allergy, an eosinophilic phase characterized by peripheral eosinophilia possibly with evidence of eosinophilic inflammation in tissues, and a third vasculitic phase in which all EGPA manifestations may occur [14]. Whether EGPA has specific triggering factors is unknown. Some studies demonstrated the presence of clonal TCR (T-cell receptor) expansion of CD8+ T cells in EGPA patients, but not in asthmatic patients or healthy controls, suggesting that there might be an immune reaction to a specific antigenic trigger in this condition [15,16]

To the best of our knowledge, this is the fourth report providing a potential link between EGPA and anti-SARS-CoV-2 vaccination. In detail, two patients have been described having developed new-onset EGPA following vaccination (one with the mRNA-1273 vaccine, the other with the BNT162b2 vaccine) [17,18], while one patient was described experiencing an exacerbation of pre-existing EGPA in remission (following the BNT162b2 vaccine) [19]. All these patients had a previous history of asthma, developed symptoms within two weeks from vaccination, had elevated eosinophil counts, tested positive for pANCA (anti myeloperoxidase), and had multi-organ involvement. New-onset disease occurred after the second vaccine dose, whereas the EGPA flare was reported after the first dose, although the patient had experienced SARS-CoV-2 infection a few months earlier. 

All cases, including ours, occurred following an mRNA vaccine. A case-based review analyzing 29 cases of ANCA-associated vasculitis (AAV) arising in concomitance of anti-SARS-CoV-2 vaccines showed that 75% of patients developed the condition following mRNA vaccines, and that pANCA was most frequently identified [20]. Interestingly, a study on a patient who developed AAV following mRNA-based influenza vaccination showed enhanced in vitro ANCA production upon exposure to RNA-based influenza and rabies vaccines, but not for other influenza vaccines or when mRNA-based vaccines were previously treated with ribonuclease or toll-like receptor (TLR) 7 antagonists [21]. Rare cases of AAV have also been reported during or soon after SARS-CoV-2 and influenza infections [22,23,24]. Furthermore, there is still debate as to which component (i.e.,: mRNA constructs or nanocarriers) is responsible for immunologically mediated adverse reactions such as those from hypersensitivity [25].

Our case developed after the booster dose with an mRNA vaccine, while the first two doses were of a viral vector-based vaccine; it could be assumed that the use of different vaccine products has favored the onset of vasculitis, however, to the best of our knowledge, there is no evidence that the use of different vaccine products for standard and booster doses is associated with an increase in adverse events, including the occurrence of vasculitis. 

Notably, anti-SARS-CoV-2 vaccines are highly safe and effective, also in patients with IMDs. A study that evaluated anti-SARS-CoV-2 vaccination safety in a cohort of 686 individuals with IMDs showed that post-vaccination disease activity remained stable in the vast majority of cases, and that the prevalence of adverse events was similar between the IMD group and the healthy control group [26]. When looking specifically at patients with ANCA-associated vasculitis, disease exacerbation following vaccination is uncommon [26]. Nevertheless, in our patient, it appeared that the vaccine promoted EGPA onset, with significant and persistent worsening of asthma following the first two doses, and EGPA development following the booster dose. 

It would be tempting to speculate that RNA viruses and RNA-based vaccines can act as triggers for ANCA-associated vasculitis in predisposed individuals. However, these events are extremely rare and further studies are required to draw reliable conclusions. It appears that our patient never developed COVID-19, and we do not know if SARS-CoV-2 infection would have been able to trigger EGPA in this case. At the beginning of the COVID-19 pandemic, great concern rose over possible greater severity of infection in patients with asthma, but this was not confirmed in practice [27,28,29] and apart from rare reports there is no strong evidence suggesting that COVID-19 can be an EGPA trigger in patients with asthma. It is impossible to know if our patient would otherwise have developed EGPA at some point in his life, but our case report may suggest that caution should be paid with the administration of additional vaccine doses in individuals who experience an increase in IMD severity that persists over time following previous vaccine shots. 

## Figures and Tables

**Figure 1 vaccines-10-00716-f001:**
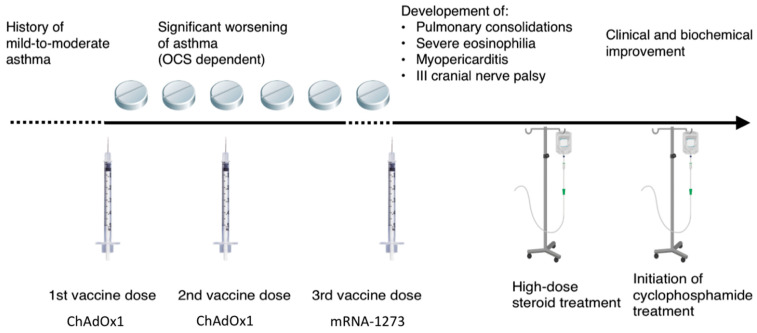
Timeline of clinical events.

**Figure 2 vaccines-10-00716-f002:**
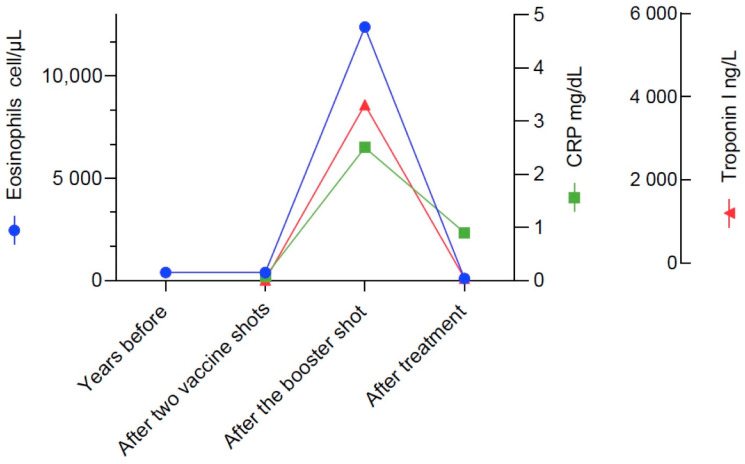
Blood parameters of disease activity in our patient at four timepoints. Notably, blood eosinophils were found to be within the normal range some years previously (in 2018) and also during the worsening of asthma which occurred after the first two vaccine shots and apparently were elevated only once EGPA was manifest.

**Figure 3 vaccines-10-00716-f003:**
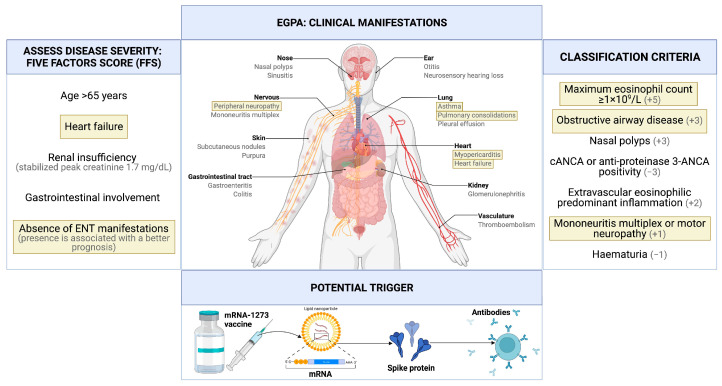
Graphical representation of clinical manifestations, items of the Five Factors Score (FFS) and ACR/EULAR classification criteria present in the described clinical case (highlighted in yellow).

## Data Availability

Not applicable.

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
