# Peer review of "New Onset of Eosinophilic Granulomatosis with Polyangiitis Following mRNA-Based COVID-19 Vaccine"

_vaccines, 2022, doi:10.3390/vaccines10050716_

Round 1

Reviewer 1 Report

In this manuscript, Enrico Heffler and his colleagues reported a case of EGPA after mRNA-based SARS-CoV-2 vaccination. The authors described the patient’s medical history, clinical characteristics, treatment, and clinical response. There have been several EGPA reports following SARS-CoV-2 mRNA vaccine injection. However, the relevance between EGPA and mRNA-based vaccination needs more cases and further studies. This study added a case into the pool. Although it is still unclear whether the mRNA vaccine triggers EGPA, the study suggests caution for individuals with IMDs.

Minor comments:

In Fig. 1, labelling the time points of vaccination and intervention would be more apparent.

Author Response

REVIEWER: In this manuscript, Enrico Heffler and his colleagues reported a case of EGPA after mRNA-based SARS-CoV-2 vaccination. The authors described the patient’s medical history, clinical characteristics, treatment, and clinical response. There have been several EGPA reports following SARS-CoV-2 mRNA vaccine injection. However, the relevance between EGPA and mRNA-based vaccination needs more cases and further studies. This study added a case into the pool. Although it is still unclear whether the mRNA vaccine triggers EGPA, the study suggests caution for individuals with IMDs.

RESPONSE: We thank the Reviewer for his/her nice comments. We also believe that more cases and furthers studies about the relation between SARS-CoV-2 vaccination and new onset of EGPA are needed, and we thank again the Reviewer for considering our case a piece of this complex puzzle.

REVIEWER:  In Fig. 1, labelling the time points of vaccination and intervention would be more apparent.

RESPONSE: In the revised version of our manuscript, Figure 1 was modified according to the Reviewer's suggestions.

Reviewer 2 Report

The case report entitles “New onset of Eosinophilic Granulomatosis with Polyangiitis 2 following mRNA-based COVID-19 vaccine” was submitted by Nappi et al to the “Vaccines, ID 1708311)” clearly articulates the onset of Eosinophilic Granulomatosis with Polyangiitis following mRNA-based COVID-19 vaccine. Abstract: Anti-SARS-CoV-2 vaccines are safe and effective, also in individuals with allergic and 15 immune-mediated diseases (IMDs). There are reports suggesting that vaccines may be able to 16 trigger de-novo or exacerbate pre-existing IMDs in predisposed individuals. Eosinophilic granu-17 lomatosis with polyangiitis (EGPA) is a small vessel vasculitis characterized by asthma, eosino-18 philia, and eosinophil-rich granulomatous inflammation in various tissues. We describe the case of 19 a 63-year-old man who experienced cardiac, pulmonary, and neurological involvement one day 20 after the administration of the third dose of anti-SARS-CoV-2 vaccine (mRNA-1273). A diagnosis of 21 EGPA was made and the patient was treated with high-dose steroids and cyclophosphamide, with 22 a good clinical response. Interestingly, our patient experienced a significant worsening of his 23 pre-existing asthma six months earlier, just after the first two vaccine shots with the ChAdOx1 an-24 ti-SARS-CoV-2 vaccine. It is impossible to know whether our patient would have had developed 25 EGPA following natural SARS-CoV-2 infection or at some point in his life regardless of infectious 26 stimuli. Nevertheless, our report may suggest that caution should be paid with the administration 27 of additional vaccine doses in individuals that experiences an increase in IMD severity that persists 28 over time following previous vaccine shots. 

However, there are certain major questions for the authors as follows:

The patient has administrated by first two doses with the ChAdOx1 and booster shots with mRNA-1273. It is well known that the ChAdOx1 is the viral vector-based vaccine and mRNA-1273 is RNA based vaccine. Both are two different platforms.

How did the authors confirm, that the EGPA or worsening of asthma after vaccination, was caused by the mRNA vaccine?

Does the mixing of vaccines cause EGPA in the particular case?. The authors need to clarify.

The author may refer to and cite the https://doi.org/10.1007/s12539-021-00438-3

Also, the present report shows, after the initiation of systemic corticosteroid therapy, observed depletion of blood eosinophils, progressive reduction of serum troponin, resolution of cough, and almost complete regression of cranial nerve palsy. The authors need to discuss, whether the patient suffered from any mild to adverse reactions including indigestion or hiccups, during the steroid treatment period?.

........................................................................................................................................

Author Response

REVIEWER: The case report entitles “New onset of Eosinophilic Granulomatosis with Polyangiitis 2 following mRNA-based COVID-19 vaccine” was submitted by Nappi et al to the “Vaccines, ID 1708311)” clearly articulates the onset of Eosinophilic Granulomatosis with Polyangiitis following mRNA-based COVID-19 vaccine. 

RESPONSE: We sincerely thank the Reviewer for his/her nice comment.

REVIEWER: The patient has administrated by first two doses with the ChAdOx1 and booster shots with mRNA-1273. It is well known that the ChAdOx1 is the viral vector-based vaccine and mRNA-1273 is RNA based vaccine. Both are two different platforms. How did the authors confirm, that the EGPA or worsening of asthma after vaccination, was caused by the mRNA vaccine?

RESPONSE: This is a very good point. We concluded as probable mRNA vaccine-induced EGPA as systemic vasculitic clinical manifestations arised after the administration of mRNA-1273, while ChAdOx1 admnistration was associated with worsening of asthma. It is likely that the vaccination stimulus itself was able to trigger an immune response such as to lead to worsening of asthma first, and subsequently to systemic vasculitis. We reported this comment in the Discussion of the revised manuscript.

REVIEWER: Does the mixing of vaccines cause EGPA in the particular case?. The authors need to clarify.

RESPONSE: Thank you for this very interesting question: so far, to our knowledge, there is no evidence that mixing of vaccines can be detrimental in terms of increased risk of adverse events, including new onset of vasculitis. We added this comment in the Discussion of the revised manuscript.

REVIEWER: The author may refer to and cite the https://doi.org/10.1007/s12539-021-00438-3

RESPONSE: Thank you very much for reporting this very recent reference that we have cited in the revised version of the manuscript

REVIEWER: Also, the present report shows, after the initiation of systemic corticosteroid therapy, observed depletion of blood eosinophils, progressive reduction of serum troponin, resolution of cough, and almost complete regression of cranial nerve palsy. The authors need to discuss, whether the patient suffered from any mild to adverse reactions including indigestion or hiccups, during the steroid treatment period?

RESPONSE: to date, more than 3 months after starting high-dose systemic corticosteroid therapy, the patient has not reported any adverse reaction (even mild ones)

Round 2

Reviewer 2 Report

Accept in present form.